# The repositioning of epigenetic probes/ inhibitors identifies new anti-schistosomal lead compounds and chemotherapeutic targets

**Kezia C. L. Whatley**[1☉], **Gilda Padalino**[1☉], **Helen Whiteland**[1], **Kathrin K. Geyer**[1], **Benjamin J. Hulme**[1], **Iain W. Chalmers**[1], **Josephine Forde-Thomas**[1], **Salvatore Ferla**[2], **Andrea Brancale**[2], **Karl F. Hoffmann**[1]*

**1** Institute of Biological, Environmental and Rural Sciences (IBERS), Aberystwyth University, Aberystwyth, United Kingdom, **2** School of Pharmacy and Pharmaceutical Sciences, Cardiff University, King Edward VII Avenue, Cardiff, United Kingdom

☉ These authors contributed equally to this work.
* krh@aber.ac.uk

**Data Availability Statement:** All relevant data are within the manuscript and its Supporting Information files.

## Abstract

### Background

Praziquantel represents the frontline chemotherapy used to treat schistosomiasis, a neglected tropical disease (NTD) caused by infection with macro-parasitic blood fluke schistosomes. While this drug is safe, its inability to kill all schistosome lifecycle stages within the human host often requires repeat treatments. This limitation, amongst others, has led to the search for novel anti-schistosome replacement or combinatorial chemotherapies. Here, we describe a repositioning strategy to assess the anthelmintic activity of epigenetic probes/ inhibitors obtained from the Structural Genomics Consortium.

### Methodology/Principle findings

Thirty-seven epigenetic probes/inhibitors targeting histone readers, writers and erasers were initially screened against *Schistosoma mansoni* schistosomula using the high-through-put Roboworm platform. At 10 µM, 14 of these 37 compounds (38%) negatively affected schistosomula motility and phenotype after 72 hours of continuous co-incubation. Subsequent dose-response titrations against schistosomula and adult worms revealed epigenetic probes targeting one reader (NVS-CECR2-1), one writer (LLY-507 and BAY-598) and one eraser (GSK-J4) to be particularly active. As LLY-507/BAY-598 (SMYD2 histone methyl-transferase inhibitors) and GSK-J4 (a JMJD3 histone demethylase inhibitor) regulate an epigenetic process (protein methylation) known to be critical for schistosome development, further characterisation of these compounds/putative targets was performed. RNA interference (RNAi) of one putative LLY-507/BAY-598 *S. mansoni* target (Smp_000700) in adult worms replicated the compound-mediated motility and egg production defects. Furthermore, H3K36me2, a known product catalysed by SMYD2 activity, was also reduced by LLY-507 (25%), BAY-598 (23%) and si*Smp_000700* (15%) treatment of adult worms.

**Funding:** AB and SF acknowledge the support of the Life Science Research Network Wales grant no. NRNPGSep14008, an initiative funded through the Welsh Government's Sêr Cymru program. SF is supported by the Sêr Cymru II programme which is part-funded by Cardiff University and the European Regional Development Fund through the Welsh Government. The SGC receives funds from AbbVie, Bayer, Boehringer Ingelheim, the Canada Foundation for Innovation, the Canadian Institutes for Health Research, Genome Canada, GlaxoSmithKline, Janssen, Lilly Canada, the Novartis Research Foundation, the Ontario Ministry of Economic Development and Innovation, Pfizer, Takeda, and the Wellcome Trust (092809/Z/10/Z). This work was additionally supported by a Wellcome Trust grant (107475/Z/15/Z) awarded to KFH. The funders had no role in study design, data collection and analysis, decision to publish, or preparation of the manuscript.

**Competing interests:** The authors have declared that no competing interests exist.

Oviposition and packaging of vitelline cells into *in vitro* laid eggs was also significantly affected by GSK-J4 (putative cell permeable prodrug inhibitor of Smp_034000), but not by the related structural analogue GSK-J1 (cell impermeable inhibitor).

## Conclusion/Significance

Collectively, these results provide further support for the development of next-generation drugs targeting schistosome epigenetic pathway components. In particular, the progression of histone methylation/demethylation modulators presents a tractable strategy for anti-schistosomal control.

## Author summary

Human schistosomiasis is caused by infection with parasitic blood fluke worms. Global control of this NTD is currently facilitated by administration of a single drug, praziquantel (PZQ). This mono-chemotherapeutic strategy of schistosomiasis control presents challenges as PZQ is not active against all human-dwelling schistosome lifecycle stages and the evolution of PZQ resistant parasites remains a threat. Therefore, new drugs to be used in combination with or in replacement of PZQ are urgently needed. Here, continuing our studies on *Schistosoma mansoni* epigenetic processes, we performed anthelmintic screening of 37 epigenetic probes/epigenetic inhibitors obtained from the Structural Genomics Consortium (SGC). The results of these studies highlighted that schistosome protein methylation/demethylation processes are acutely vulnerable. In particular, compounds affecting schistosome SMYD (LLY-507, BAY-598) or JMJD (GSK-J4) homologues are especially active on schistosomula and adult worms during *in vitro* phenotypic drug screens. The active epigenetic probes identified here as well as their corresponding *S. mansoni* protein targets offers new starting points for the development of next-generation anti-schistosomals.

## Introduction

Amongst human infectious diseases caused by macro-parasitic organisms, schistosomiasis is the most significant in terms of its negative impact on both individual health and population-driven socio-economic outputs [1–3]. The current cornerstone of schistosomiasis control in endemic communities is preventative chemotherapy with praziquantel (PZQ), a pyrazinoiso-quinoline-like compound that induces minimal side effects and demonstrates a highly-favourable absorption, distribution, metabolism and excretion (ADME) profile [4]. However, as PZQ has been the primary anti-schistosomal used across the globe for the past three decades [5] and its currently unknown mechanism of action (possibly modulating serotonin signalling; [6, 7]) is variably effective against intra-human schistosome lifecycle stages [8], the search for PZQ replacement or combinatorial drugs is under intense investigation should drug resistant schistosomes develop.

One recent approach applied to schistosome drug discovery is based on the concept of compound repositioning or repurposing, where new indications for existing drugs are sought [9]. Two benefits of such a repositioning strategy for schistosomiasis include: 1) accelerating the drug discovery pipeline due to pre-existing safety and ADME data being available for the

repositioned compound and 2) identifying putative anti-schistosome candidate proteins due to target-based aspects of pharmaceutical-led, drug developmental programmes [10]. Sourcing of repositionable compounds, from industrial suppliers, for use in academic laboratories engaged in anthelmintic research has been recently facilitated by the efforts of not—for—profit organisations including the World Intellectual Property Organisation Re:Search—BioVentures for Global Health consortium [11] and the Structural Genomics Consortium (SGC) [12]. It is expected that these academic/industry private—public—partnerships will build upon previous repositioning successes in the identification of new leads for treating schistosomiasis [13–16].

Due to our (and other research groups) continued interest in deciphering how schistosome epigenetic processes shape schistosome lifecycle progression [17–27] and the SGC's ability to supply epigenetic probes (EPs)/epigenetic inhibitors (EIs) (compounds that have been designed to modulate human epigenetic targets) [28, 29], we herein have conducted a repositioning campaign focused on the anti-schistosomal activity (against *S. mansoni*) of thirty-four SGC-supplied EPs (compounds that display *in vitro* potency of < 100 nM, > 30 fold selectivity vs other subfamilies and on-target cellular activity at 1 μM) and three EIs (compounds that do not display these specific epigenetic probe traits) [30]. Using both a high-throughput platform for measuring schistosomula motility and phenotype as well as a low-throughput assay for quantifying adult schistosome motility and egg production [31–33], we have determined that compounds targeting bromodomain (BRD)–containing proteins, histone methyltransferases (HMTs) and histone demethylases (HDMs) are amongst the most potent anti-schistosomals within the tested SGC epigenetic probe collection. As the schistosome histone methylation machinery has recently been shown to be critical for developmental processes including egg production, miracidium to sporocyst transformation and adult worm motility [34–36], we specifically pursued the SGC epigenetic probes (LLY-507/BAY-598 and GSK-J4) involved in HMT/HDM inhibition for follow-on functional investigations.

RNA interference (RNAi) and molecular modelling methods were used to functionally validate the most likely *S. mansoni* target (Smp_000700) of the HMT inhibitors LLY-507 [37] and BAY-598 [38]. Here, drug treatment or RNAi of *smp_000700* in adult worms both led to decreases in dimethylated (me2) H3K36 (Histone 3, Lysine 36), a known substrate of SMYD activity. Furthermore, inhibition of the most likely *S. mansoni* target (Smp_034000) by cell permeable GSK-J4 (but not GSK-J1) [39] led to egg production deficiencies and vitellocyte packaging defects when adult worm pairs were co-cultured with this compound at concentrations as low as 390 nM. As such, these particular epigenetic probes and their corresponding molecular targets represent exciting new leads in the identification and development of next generation anthelmintics for combating a major NTD of low to middle income countries.

## Materials and methods

### Ethics statement

All procedures performed on mice adhered to the United Kingdom Home Office Animals (Scientific Procedures) Act of 1986 (project license PPL 40/3700) as well as the European Union Animals Directive 2010/63/EU and were approved by Aberystwyth University's Animal Welfare and Ethical Review Body.

### Compound acquisition, storage and handling

All epigenetic probes, epigenetic inhibitors and chemotype matched controls (where available) were received from the SGC and solubilised in DMSO (Fisher Scientific, Loughborough, UK) at stock aliquot concentrations of 1.6 mM and 10 mM for schistosomula and adult worm screening respectively. Auranofin (AUR) and praziquantel (PZQ) were purchased from

Sigma-Aldrich and reconstituted similarly to the SGC compounds. All reconsited compounds were stored at -80˚C.

## Parasite material

A Puerto Rican strain (NMRI) of *S. mansoni* was used throughout the study and passaged between *Mus musculus* (Tuck Ordinary; TO) and *Biomphalaria glabrata* (NMRI albino and pigmented hybrid [40]) hosts. Cercariae were shed from both *B. glabrata* strains by exposure to light in an artificially heated room (26˚C) for 1 h and used to percutaneously infect *M. musculus* (200 cercariae/mouse) [41] for generation of adult schistosomes or to mechanically transform into schistosomula [42] for *in vitro* compound screening. Adult schistosomes were obtained from *M. musculus* at 7 wks post-infection and used for *in vitro* compound screening and RNA interference (RNAi).

## Bioinformatics

The specific *Homo sapiens* target (histone reader, writer or eraser) of the SGC-provided epigenetic probes/inhibitors initially was derived from the SGC website and corresponding reference literature [30]. Uniprot IDs of the representative *H. sapiens* epigenetic target were obtained from Uniprot [43] and their downloaded protein sequences were used as queries for protein BLAST (BLASTp) searches against the predicted protein database derived from the *S. mansoni* genome hosted in NCBI [44] and Wormbase-Parasite [45] using default settings. For the BLASTp searches, both full-length and catalytic domain peptide sequences were used as queries against the *S. mansoni* genome (v7.0). The most closely related *S. mansoni* protein (as compared by *E* values) and their sequence similarity were reported. Multiple sequence alignments (MSAs) of the amino acid sequence of the catalytic domain amino acid sequences within the identified SmSMYD and human SMYD proteins were performed with MUSCLE v3.8 (Multiple Sequence Comparison by Log Expectation) using the default parameters [46] and selecting ClustalW as the output format [47]. The alignments were then analysed in Jalview v2.9 [48] and visually inspected to check for ambiguities and sequences not aligning correctly. Phylogenetic trees were constructed by MEGA7 using the neighbour-joining method based on the JTT matrix-based model with default settings [49]. A total of 1000 bootstrap replications were run to estimate the confidence of each node.

## *In vitro* schistosomula screening

Mechanically transformed schistosomula phenotype and motility metrics were assessed at 72 h post compound incubation as previously described [50], with minor modifications. Briefly, 384-well tissue culture plates (Perkin Elmer, cat 6007460), containing 20 µl of Basch medium [51], were wet-stamped using a Biomek NX$^P$ liquid handling platform (Beckman Coulter, UK) with negative (0.625% dimethyl sulfoxide, DMSO) and positive controls (AUR and PZQ at 10 µM final concentration in 0.625% DMSO) as well as 37 SGC compounds (at 10 µM final concentration in 0.625% DMSO). Two-fold titrations of 14 SGC compounds, which were consistent hits at 10 µM, were also conducted at 10 µM, 5 µM, 2.5 µM, 1.25 µM and 0.625 µM to generate approximate EC$_{50}$s using GraphPad Prism. To each pre-loaded well (single concentration or titration screens), a total of 100–120 mechanically transformed schistosomula (in 60 µl) were deposited via a WellMate (Thermo Scientific, UK). Schistosomula/compound co-cultures (80 µl in total) were then incubated at 37˚C for 72 h in a humidified atmosphere containing 5% $CO_2$. At 72 h, tissue culture plates (containing schistosomula/compound co-cultures) were imaged under the same conditions (37˚C in a humidified atmosphere containing 5% $CO_2$) using an ImageXpressXL high content imager (Molecular Devices, UK) with

subsequent images processed for phenotype and motility as previously reported [50] and successfully implemented by us at Aberystwyth University [7, 32–34, 52]. Single point schistosomula screens (10 μM) were repeated two or three times whereas dose response titrations were performed once. Z´ values obtained from all schistosomula screens ranged from 0.27–0.48 (mean = 0.40) for phenotype and 0.41–0.56 (mean = 0.48) for motility.

### *In vitro* adult schistosome screening

The anthelmintic effect that selected SGC compounds (i.e. those 14/37 compounds that demonstrated anti-schistosomula activity at 10 μM) had on adult male and female schistosome pairs as well as egg production was assessed according to the methodology described by Edwards *et al.* [31]. *In vitro* screening was replicated three times on different dates to account for any biological variation. Briefly three adult worm pairs/well (48-well tissue culture plate format) were placed into DMEM (Gibco, Paisley, UK) supplemented with 10% (v/v) FCS (Gibco, Paisley, UK), 1% (v/v) L-glutamine (Gibco, Paisley, UK) and an antibiotic mixture (150 Units/ml penicillin and 150 μg/ml streptomycin; Gibco, UK). DMSO (0.5% negative control), AUR (10 μM concentration in 0.5% DMSO; positive control) and two-fold titrations of SGC compounds (50 μM– 6.25 μM or 50 μM– 0.05 μM; in 0.5% DMSO max) were added, and together, these adult worm/compound co-cultures were incubated at 37˚C for 72 h in a humidified atmosphere containing 5% $CO_2$. Following compound incubation, WHO/TDR readouts of adult worm motility [53], abundance of H3K36me2 (LLY-507 and BAY-598 treated worms only) in histone extracts and egg counts/well were recorded at 72 h.

### RNA interference (RNAi)

Following the perfusion of 7 wks infected mice, adult worms were recovered and RNAi performed as previously described [20, 21]. Briefly, *smp_000700* and non-specific *luciferase* siRNA duplexes were purchased from Sigma (si*Smp_000700* = sense: GGUAAUCGGUCAUG UGUAU[dT][dT] and anti-sense: AUACACAUGACCGAUUACC[dT][dT]; si*Luc* = sense CUUACGCUGAGUACUUCGA[dT][dT] and anti-sense UCGAAGUACUCAGCGUAAG [dT][dT]) and used at a final concentration of 50 ng/μl. Mixed sex adult worm pairs (for knockdown assessment by quantitative reverse transcription PCR, qRT-PCR) were cultured at 37˚C in DMEM supplemented with 10% FCS, 2 mM L-glutamine, 10% v/v HEPES (Sigma-Aldrich, UK), 100 Units/ml penicillin and 100 μg/ml streptomycin in an atmosphere of 5% $CO_2$ with a 70% media exchange performed every 48 h. The experiment was replicated three times (21 worm pairs/replicate). Quantitative reverse transcription PCR (qRT-PCR) of *smp_000700* abundance was performed at 48 h and adult worm motility [53] as well as egg counts were quantified at 168 h. Levels of H3K36me2 detected in schistosome histone extracts were assessed at 72 h.

### Quantitative reverse transcription PCR (qRT-PCR)

Following RNAi with si*Smp_000700* and si*Luc*, mixed-sex adult worms were incubated for a total of 48 h before processing them for RNA isolation. Briefly, worms were homogenised using a TissueLyser LT (Qiagen, UK) in TRIzol Reagent (Invitrogen, UK) before isolation of total RNA using the Direct-zol RNA Kit (Zymo, UK). cDNA was then generated using the SensiFAST cDNA synthesis kit (Bioline), qRT-PCR performed and data analysed as previously described [54]. qRT-PCR primers for amplifying *smp_000700* (Forward primer 5'-GTCTTGC ATGTATAGAGGATTGGTC-3', Reverse 5'-GCAGTCAACCGATTCAATTAAAGT-3') and internal standard *alpha tubulin* (SmAT1; Forward primer 5'-CGAAGCTTGGGCGCGTCTA GAT -3', Reverse 5'-CTAATACTCTTCACCTTCCCCT -3') were purchased from Sigma.

## Homology modelling and epi-drug docking

Homology modelling of Smp_000700 was prepared as previously described [34] with some minor modifications. The homology model of full length Smp_000700 was constructed using the crystal structure of the human SMYD3 (PDB ID: 5EX3 [55]) as a template containing a substrate peptide and S-adenosyl homocysteine (SAH), the demethylated metabolite of the cofactor S-adenosyl methionine (SAM). Although the sequence similarity between the parasite target and the homologous human protein is not very high (32%), it's still above the critical level (30%) in producing homology models [56].

The FASTA amino acid sequences for the protein to be modelled was obtained from Uniprot [57] and then used to perform a protein BLAST (BLASTp) search in NCBI [44] to obtain a homologous sequence of a protein to be used as a template. The crystal structure of the selected protein (*H. sapiens* SMYD3, PDB ID: 5EX3) suitable as a template was downloaded from PDB. The homology model was built using the homology modelling tool and a single template approach with Amber99 force field in MOE2018.10 [58]. Briefly, the primary sequence of Smp_000700 to be modelled was loaded into MOE together with the 3D structure of HsSMYD3, both sequences were aligned and the final 3D model of Smp_000700 was produced as a single output structure. The model was subsequently refined by energy minimization with RMSD of 0.1. MOE-Homology (developed by Chemical Computing Group, Inc.) combines the methods of segment-matching procedure and the approach to the modelling of insertion/deletion regions [59]. Using default parameters, four different softwares (Ramachandran plot analyses, ProSA-web, ERRAT and Verify3D) were used to validate the robustness of the homology model [60–63].

Docking simulations of BAY-598 and LLY-507 were performed using the Glide docking software within Maestro (Schrödinger Release 2017 [64]) as previously discussed [34]. The model was pre-processed using the Schrödinger Protein Preparation Wizard by assigning bond orders, adding hydrogens and performing a restrained energy minimisation of the added hydrogens using the OPLS_2005 force field. The docking site was identified over the substrate binding pocket of each homology model and a 12 Å docking grid (inner-box 10 Å and outer-box 22 Å) was prepared using, as a centroid, the substrate peptide. Glide SP precision was used keeping the default parameters and setting 5 as number of output poses per input ligand to include in the solution. The output poses were saved as a mol2 file. The docking results were visually inspected for their ability to bind the active site.

## H3K36me2 detection

LLY-507 (6.25 μM), BAY-598 (25 μM) or si*Smp_000700* treated male and female worms (alongside DMSO or si*Luc* controls; 21 individuals per condition) were homogenized with a TissueLyser (Qiagen) and total histones extracted using the EpiQuik[TM] Total Histone Extraction Kit (Epigentek). Levels of H3K36me2 in adult histone extracts were measured using the EpiQuik[TM] Global Di-Methyl Histone H3-K36 Quantification Kit (Fluorimetric, Epigentek). Technical duplicates of three biological replicates of each treatment were analysed according to the manufacturer's instructions. Fluorescent readings ($530_{EX}/590_{EM}$ nm) were obtained using a POLARstar Omega (BMG Labtech, UK) microtiter plate reader. Fluorescent values of the samples were corrected by subtracting the fluorescent readings of the blank (buffer only, provided in the kit). The mean of the adjusted control values (DMSO for LLY-507/BAY-598 treated worms and si*Luc* for si*Smp_000700* treated worms) was set at 100% H3K36me2 and the standard deviation (SD) was calculated from the normalised values.

## Vitellocyte and egg volume quantification

Prior to laser scanning confocal microscopy (LSCM) visualisation, the total number of eggs produced by GSK-J4 (0.2 μM), GSK-J1 (6.25 μM) or DMSO treated adult worm pairs were enumerated and subsequently immersed in PBS supplemented with DAPI (4',6-diamidino-2-phenylindole, 2 μg/ml). Fluorescent microscopic images (10 eggs per treatment) were captured on a Leica TCS SP8 super resolution laser confocal microscope fitted with a 63 X (water immersion) objective using the Leica Application Suite X. Green (egg autofluorescence) fluorescence was visualised with an argon or diode-pumped, solid state (DPSS) laser at 488 nm. DAPI was visualised using a 405 nm blue diode laser. The number of vitellocytes (DAPI$^+$ cells) and overall volume (mapped by the green autofluorescence) for individual eggs were calculated using IMARIS 7.3 software (Bitplane).

## HepG2 cell culture and MTT assays

Human Caucasian Hepatocyte Carcinoma (HepG2) cells (Sigma Aldrich, UK) were utilized to assess SGC epigenetic probe cell cytotoxicity in the application of the MTT (3-(4,5-dimethylthiazol-2-yl)-2,5-diphenyltetrazolium bromide) assay [52]. Briefly, cells were passaged at 70–80% confluence and seeded (20,000 cells/ 50 μL per well) in a 96-well, black-sided, clear-bottomed falcon plate (Fisher Scientific, UK) with the final plate column (8 wells) being treated as a blank (media only). Following a 24 h incubation (5% $CO_2$, 37˚C, humidified), HepG2 cells were then treated with 50 μL of pre-warmed media (37˚C) containing SGC EPs/EIs at 200 μM to 0.02 μM (final concentration 100 μM, 10 μM, 1 μM, 0.1 μM and 0.01 μM). Three positive (1% Triton X-100) and negative (1% DMSO) control wells per plate were additionally included.

After addition of compounds, each plate was then incubated for a further 20 h before application of MTT for assessment of overt compound cytotoxicity using the MTT assay [52, 65]. The MTT assay was read using the POLARstar Omega (BMG LabTech, UK) plate reader at an absorbance of 570 nm. $CC_{50}$s were calculated in GraphPad Prism.

## Statistical analysis

All statistical analyses were performed using a Student's *t*-test (two samples) or a two-way ANOVA followed by Least Significant Difference post-hoc correction (more than two samples). A *p* value less than 0.05 was considered statistically significant.

# Results and discussion

## Compounds obtained from the Structural Genomics Consortium (SGC) display anti-schistosomal activity

Continuing our search for novel anti-schistosomal drug targets from within the parasite's epigenetic machinery has led to the acquisition of 34 EPs and 3 EIs targeting histone modifying enzyme (HME) readers, writers and erasers from the SGC (S1 Table). The difference between EPs and EIs is based on the compound's *in vitro* activity (EPs display *in vitro* potency > 100 nM; EIs do not), selectivity (EPs display > 30-fold selectivity vs other subfamilies; EIs do not) and on-target cell activity (EPs display activity > 1 μM; EIs do not) [30]. These EPs and EIs have initially been developed for oncology research and inflammatory disorders in humans or animal models [30]. Because of this, the human target is well characterised allowing us to identify the most likely *S. mansoni* target of each of these EPs and EIs by BLASTp analysis of the parasite's genome (Table 1).

In most cases, a clear one to one *S. mansoni* homolog (*E* values less than $1E^{-10}$) could be identified for all EP and EI targets. However, no support for a schistosome PAD-4 (Uniprot

Q9UM07) homolog was found in the *S. mansoni* genome (v7.0) and only weak support for a SMYD2 (Uniprot Q9NRG4) protein lysine methyltransferase (PKMT) homolog could be identified. In this later case, both Smp_342100 and Smp_000700 contained moderate sequence similarity (over the full-length or main catalytic domain sequences) to HsSMYD2 suggesting that this particular SMYD homolog may present unique (non-human) features useful from a drug development standpoint. In cases where EPs or EIs are unable to distinguish between family members such as PFI-1, JQ1 and BSP antagonists targeting the BET (Bromo and Extra-Terminal domain protein) family, IOX1 targeting the JMJD (JuMonJi C Domain-containing) protein demethylase family or LAQ824 and CI-994 targeting the class I HDACs (Histone DeA-Cetylases), homologous schistosome families were the presumed targets. Therefore, a single Smp could not be identified for these compounds. Following on from this comparative sequence analysis, a single-point (10 μM) schistosomula screen of these 37 compounds was implemented to assess their anthelmintic properties (Table 1 and Fig 1).

Using the Roboworm platform [52], 13 EPs and 1 EI (LAQ824) were classified as hits (14/37; 38% hit rate) affecting both schistosomula phenotype (Fig 1A) and motility (Fig 1B) at 10 μM. These hits consisted of 5 compounds targeting epigenetic readers, 6 targeting epigenetic writers and 3 targeting epigenetic erasers. Reassuringly, GSK343 was identified as a hit in these screens, supporting previous anti-schistosomal investigations [66] and providing further confidence in the results obtained for the additional EPs and EIs. In one (out of three) JQ1 replicate screen, the motility result was not considered a hit as the value fell above the 'hit' cut-off (-0.35). However, as the other two replicates were identified as motility hits and all three replicates were within the phenotype hit threshold (below -0.15), JQ1 was included as an anti-schistosomal EP. Roboworm 'hit' cut-off boundaries are fully defined by Paveley *et al* [50] and have been successfully applied in our laboratory during other anthelmintic projects [7, 32–34, 52]. Furthermore, not all EPs targeting the same protein (or family) were equally active in these studies (Table 1 and Fig 1) [30]. For example, while JQ1 was a hit, PFI-1 and BSP were not; these compounds are all promiscuous BET bromodomain inhibitors [67–69]. Similarly, PFI-4 was a hit, but the promiscuous EPs NI-57 and OF-1 were not; these compounds all target human BRPF1/2/3 (BRomodomain and PHD Finger containing) members [70]. Likewise, SGC-CBP30 demonstrated activity against schistosomula, but I-CBP112 did not; both compounds were developed against the bromodomains within human CREBBP (cAMP Responsive Element Binding protein Binding Protein) and EP300 transcriptional co-activators [71, 72]. Finally, the BRomoDomain-9 epigenetic probe I-BRD9 [73], but not LP99 [74] and BI-9564 [75] was active against schistosomula. These results, and others including the pan-acting HDAC inhibitors LAQ824 (hit) vs CI-994 (non-hit), would suggest that not all EPs designed against the same human targets (or families) are equally effective against *S. mansoni* homologues. Two exceptions to this interpretation exist as both GSK343 [76] and UNC1999 [77] (EZH1/H2 EPs) as well as BAY-598 [38] and LLY-507 [37] (SMYD2 EPs) are hits against schistosomula. Therefore, in these cases, the use of complementary EPs may provide stronger support for schistosome target validation and progression during further investigations.

Microscopic assessment of affected schistosomula demonstrated a range of abnormal phenotypes including granulation, swelling, elongation and other irregular shape modifications (Fig 1C). Some compounds (NVS-CECR2-1, LLY-507 and GSK-J4) induced phenotypes comparable to auranofin. Where chemotype-matched negative (or less active) control compounds for hits were available (A-197 for A-196, BAY-369 for BAY-598, UNC2400 for UNC1999, GSK-J1/GSKJ5 for GSK-J4), they were also tested against schistosomula at 10 μM (S1 Fig). None of these chemotype controls demonstrated anti-schistosomula activity suggesting that specificity of the hits for the *S. mansoni* homologue is similar to that found for the original human target. Further SAR of the chemotype-matched controls and their related hit EPs/EIs

**Table 1. The Structural Genomics Consortium (SGC) epigenetic probes (EPs) and epigenetic inhibitors (EIs) used in this study.**

| | SGC Compound ID | *H. sapiens* target | Uniprot | *S. mansoni* Homologue (Smp_xxxxxx) ident. by BLASTp analysis | | | | Hit on schisto-somula (10µM) |
|---|---|---|---|---|---|---|---|---|
| | | | | Full-length based | *E* value | Main catalytic domain based | *E* value | |
| READERS | NI-57 | BRD1 (BRPF1/2/3) | P55201 | Smp_246920 | 7.20E-89 | Smp_246920 | 1.10E-29 | |
| | OF-1 | | O95696 | | 1.80E-88 | | 4.50E-18 | |
| | PFI-4 | | Q9ULD4 | | 7.40E-87 | | 3.00E-22 | X |
| | LP99 | BRD9/7 | Q9H8M2/Q9NPI1 | Smp_246920 | 1.20E-19 | Smp_246920 | 2.80E-18 | |
| | BI-9564 | | | | 8.30E-21 | | 2.70E-20 | |
| | I-BRD9 | BRD9 | Q9H8M2 | Smp_246920 | 1.20E-19 | Smp_246920 | 2.80E-18 | X |
| | PFI-1 | BET family** | - | - | - | - | - | |
| | JQ1 | BET family** | - | - | - | - | - | X |
| | NVS-CECR2-1 | CECR2 | Q9BXF3 | Smp_070190 | 4.10E-16 | Smp_070190 | 1.40E-15 | X |
| | GSK2801 | BAZ2A/2B | Q9UIF9/Q9UIF8 | Smp_170760 | 4.20E-37 | Smp_170760 | 4.50E-27 | |
| | BAZ2-ICR | | | Smp_147950 | 5.10E-13 | Smp_147950 | 2.10E-14 | |
| | I-CBP112 | CREBBP/EP300 | Q92793 (CREBBP) | Smp_127010**** | 1.20E-102 | Smp_127010**** | 1.30E-105 | |
| | SGC-CBP30 | | | | | | | X |
| | PFI-3 | SMARCA2/4 | P51531/P51532 | Smp_158050 | 0 | Smp_158050 | 2.10E-32 | |
| | | | | | 1.70E-164 | | 4.20E-35 | |
| | BSP | BET family** | - | - | - | - | - | |
| | UNC1215 | L3MBTL3 | Q96JM7 | Smp_159100 | 1.50E-23 | Smp_159100 | 2.10E-24 | |
| WRITERS | A-196 | SUV420H1/H2 | Q4FZB7/Q86Y97 | Smp_062530 | 7.40E-30 | Smp_062530 | 7.40E-31 | X |
| | | | | | 4.50E-15 | | 3.30E-15 | |
| | MS023 | PRMT type I (PRMT1, 2, 3, 4, 6 and 8) | Q99873/P55345/O60678/Q86X55/Q96LA8/Q9NR22 | Smp_029240 (SmPRMT1 and 8) Smp_337860 (SmPRMT3) | 1.1E-82 5.9E-23 | Smp_029240 (SmPRMT1 and 8) Smp_337860 (SmPRMT3) | 2.2E-83 2.1E-23 | X |
| | MS049 | PRMT4 and 6 | Q86X55/Q96LA8 | Smp_070340 (SmPRMT4) | 1.00E-72 6.00E-27 | Smp_070340 (SmPRMT4) | 3.90E-45 4.80E-27 | |
| | SGC707 | PRMT3 | O60678 | Smp_337860 | 5.90E-23 | Smp_337860 | 2.1E-23 | |
| | GSK591 | PRMT5 | O14744 | Smp_171150 | 2.00E-78 | Smp_171150 | 1.70E-78 | |
| | LLY-507 | SMYD2 | Q9NRG4 | Smp_342100 (Smp_000700***) | 0.0000098 (0.00046***) | Smp_342100 (Smp_000700***) | 0.0000047 (0.00022***) | X |
| | BAY-598 | | | | | | | X |
| | SGC0946 | DOT1L | Q8TEK3 | Smp_165000 | 2.60E-67 | Smp_165000 | 3.40E-67 | |
| | UNC0642 | G9a/GLP | Q96KQ7/Q9H9B1 | Smp_158310 | 3.70E-25 | Smp_158310 | 5.30E-26 | |
| | UNC0638 | | | | | | | |
| | A-366 | | | | 1.00E-25 | | 1.40E-26 | |
| | GSK343 | EZH1/H2 | Q92800/Q15910 | Smp_078900 | 2.10E-81 | Smp_078900 | 2.60E-83 | X |
| | UNC1999 | | | | 1.90E-79 | | 4.90E-81 | X |
| | (R)-PFI-2 | SETD7 | Q8WTS6 | Smp_190140 | 6.00E-11 | Smp_190140 | 6.00E-11 | |
| | C646* | EP300 | Q09472 | Smp_127010**** | 5.70E-102 | Smp_127010**** | 1.40E-104 | |
| ERASERS | GSK484 | PAD-4 | Q9UM07 | No homologue identified | - | No homologue identified | - | X |
| | GSK-J4 | JMJD3/UTX | O15054/O15550 | Smp_034000 | 2.30E-128 | Smp_034000 | 4.00E-124 | X |
| | | | | | 9.70E-137 | | 3.00E-130 | |
| | GSK-LSD1 | LSD1 | O60341 | Smp_150560 | 2.80E-31 | Smp_150560 | 1.10E-31 | |
| | IOX1 | 2-oxoglutarate oxygenase (JMJD) family** | - | - | - | - | - | |
| | LAQ824* | class I HDAC (HDAC1, 2, 3 and 8)** | - | - | - | - | - | X |
| | CI-994* | | | | | | | |

All compound structures can be found in S1 Table.

* = Not an SGC defined epigenetic probe as it does not display *in vitro* potency of < 100 nM, does not display >30-fold selectivity vs other subfamilies and does not have significant on-target cell activity at 1µM. These chemicals are classified as epigenetic inhibitors (EIs).

** Broad activity against family. Therefore, specific Smp targets are not listed.

*** Lowest sequence similarity amongst BLAST analyses. Therefore, the top two Smps are indicated.

**** Smp_127010 contains both a bromodomain (of CREBB) and a histone acetyl transferase domain (of EP300).

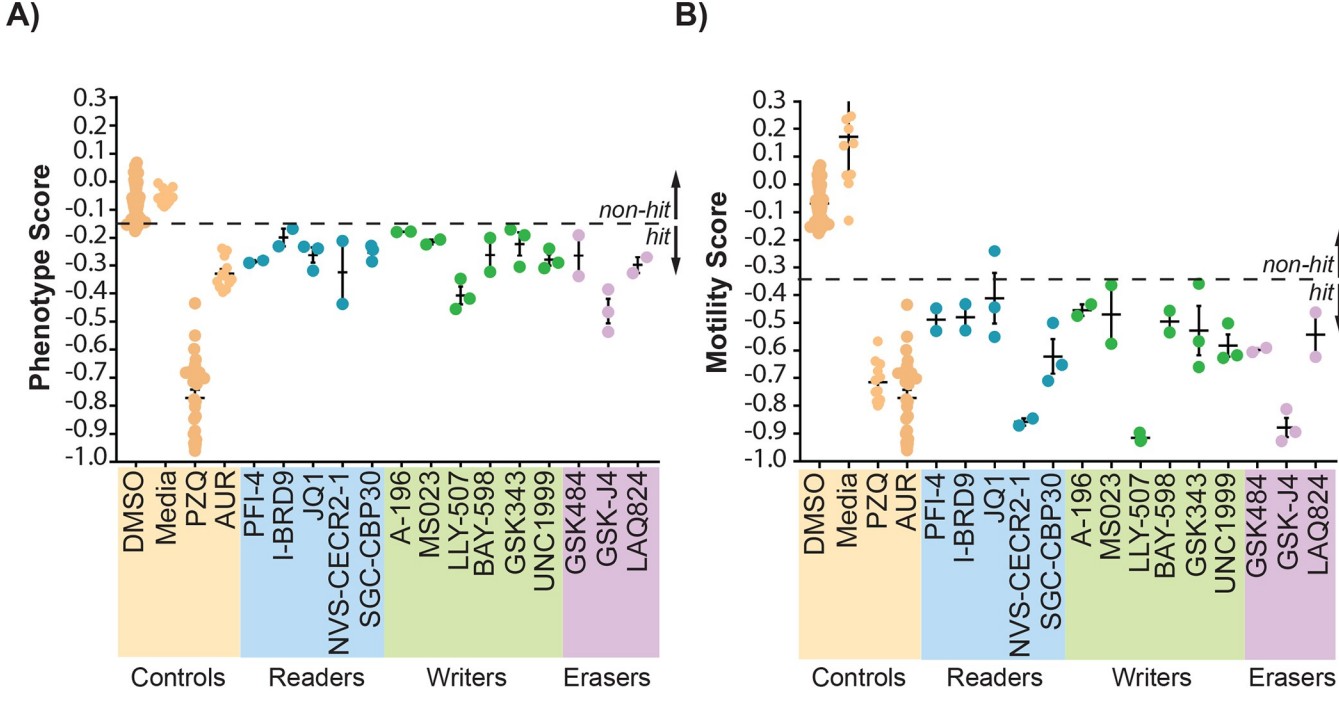

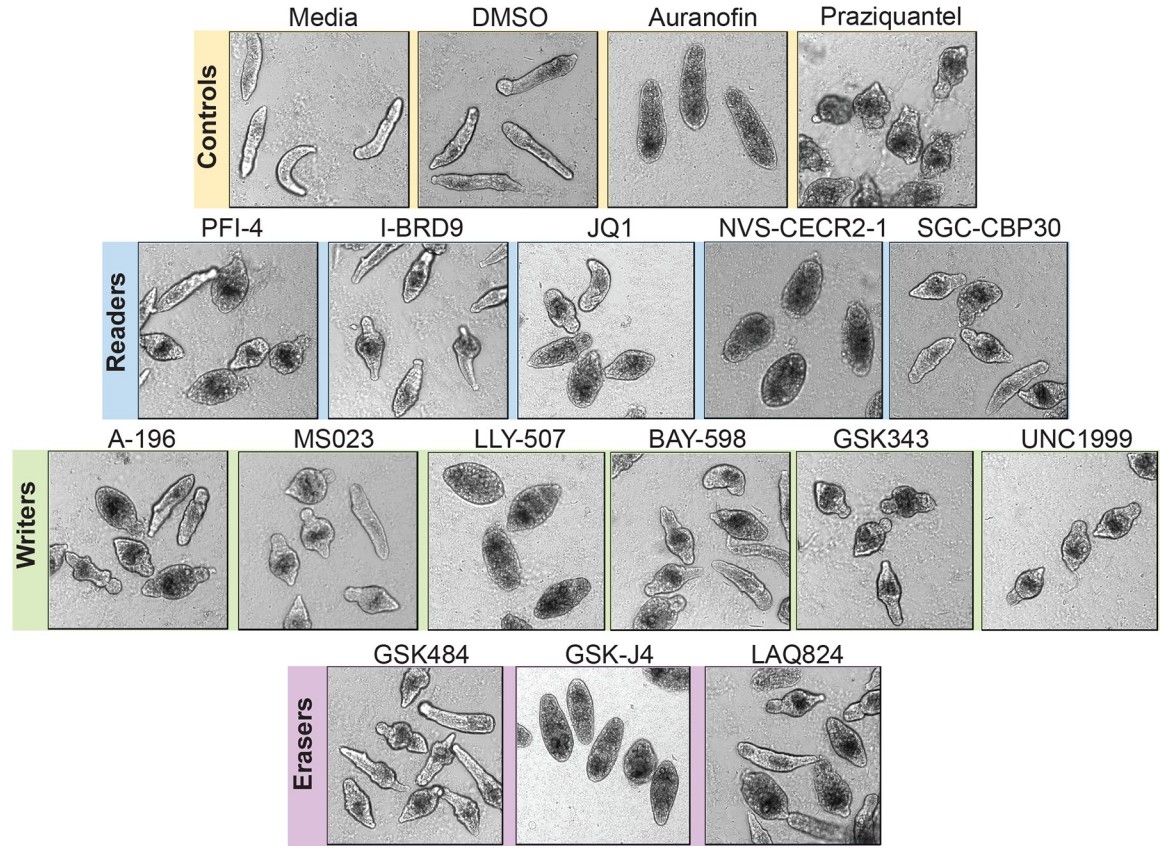

**Fig 1. Epigenetic probes/inhibitors targeting histone modifying enzymes negatively modify schistosomula phenotype and motility metrics.** A collection of 37 EPs/EIs targeting histone modifying enzymes (readers, writers and erasers) as well as controls (PZQ and AUR) were screened against *S. mansoni* schistosomula at 10 μM for 72 h using the Roboworm platform as described in the Materials and Methods. **(A)** Parasite phenotype and **(B)** Parasite motility were both negatively affected by fourteen (38%) of these compounds upon repeat screening (each filled circle represents average phenotype or motility scores derived from ~ 80–120 schistosomula; n = 2–3; horizontal bars represent average scores derived from replicates). **(C)** Representative images of schistosomula phenotypes affected by the fourteen active EPs/EIs, compared to controls (Media only, DMSO, Auranofin and Praziquantel). Z´ values obtained from these screens ranged from 0.27–0.48 (mean = 0.40) for phenotype and 0.41–0.56 (mean = 0.48) for motility. Table 1 (closest *S. mansoni* homolog) and S1 Table (structures, SMILES, MW, etc.) provide further information related to EPs/EIs screened and their putative *S. mansoni* targets.

with the schistosome target or whole organism could help provide evidence to support this contention. Nevertheless, encouraged by these single-point schistosomula screens (and the results of the chemotype controls), follow-on dose-response titrations of the 14 hits against schistosomula, adult schistosome pairs and a surrogate human cell line (HepG2) were next pursued (Table 2).

Upon dose-response titrations of these 13 EP and one EI (LAQ824) anti-schistosomula hits, several of the results warranted further comment. Firstly, several compounds had some degree of general cytotoxicity against HepG2 cells. While HepG2 cytotoxicity is a flag, it is not considered a no-go output in our compound triaging scheme. Nevertheless, the cytotoxicity results are unsurprising as these EPs were primarily developed for use in humans as anti-cancer agents [30], and, as HepG2 cells are derived from a hepatocellular carcinoma, varying sensitivity to these EPs was expected (and, as an example, confirmed for LLY-507, [37]). However, PFI-4, SGC-CBP30, A-196, BY-598 and GSK-J4 had minimal activity on this cell line's viability ($CC_{50}s > 50$ μM) in these studies. Secondly, low μM anti-schistosomula $EC_{50}$s were generally a good predictor for low μM adult worm $EC_{50}$s (e.g. NVS-CECR2-1, LLY-507, GSK-J4). However, this was not always the case (e.g. I-BRD9, SGC-CBP30, MS023) and indicated that stage specific expression of the target may be an important consideration for compound progression. Thirdly, while LLY-507 may contain some off target activity in schistosomes (similar to what has been reported for human cells [30]), the anti-schistosomal potency and selectivity of BAY-598 were similar. These results suggested that both LLY-507 and BAY-598 (likely targeting a schistosome SMYD protein lysine methyltransferase homologue, Table 1) could be exchangeable in further progression of these anti-schistosomal compounds and their predicted target (s). Finally, these dose response titrations revealed that the best anti-schistosomal hit, when considering schistosomulum and adult worm potencies alongside HepG2 cytotoxicity, was GSK-J4. Because our previous investigations have independently identified crucial developmental roles for the schistosome protein methylation machinery [34, 36], we decided to further our investigations of LLY-507/BAY-598 and GSK-J4.

## Protein lysine methyltransferase (LLY-507 and BAY-598) and demethylase (GSK-J4) modulators are amongst the most selectively-potent, anti-schistosomal epigenetic probes within the SGC collection

To provide a mechanistic context associated with the anti-schistosomal activity of LLY-507 and BAY-598, a combination of homology modelling, *in silico* molecular docking and detection of H3K36 dimethylation (me2) was performed (Fig 2).

Phylogenetic analyses of the three putative *S. mansoni* SMYD family members (Smp_121610, Smp_000700 and Smp_342100 [34]) alongside the five characterised *H. sapiens* SMYD (SMYD 1–5) proteins (using the SET domain only) indicated that *S. mansoni* does not contain a close HsSMYD2 homolog (supporting the findings in Table 1) (S2 Fig). However, as the substrate competitive inhibitors of HsSMYD2 (BAY-598 and LLY-507) are both active against schistosomula (but BAY-369, the negative control of BAY-598, is not, S1 Fig), these

**Table 2. Dose response titrations of the 14 active SGC compounds against schistosomula and adult schistosome pairs.**

| | SGC COMPOUND ID | SCHISTO-SOMULA EC$_{50}$ (µM) | | ADULT EC$_{50}$ (µM) (95% Confidence Intervals) | | CC$_{50}$ (HepG2) (µM) (95% Confidence Intervals) | SELECTIVITY INDEX | | | |
|---|---|---|---|---|---|---|---|---|---|---|
| | | Phenotype | Motility | Females | Males | | Schistosomula Phenotype | Schistosomula Motility | Adult Females | Adult Males |
| READERS | PFI-4 | 5.22 NC | 5.12 NC | 31.22 (19.11–43.32) | 30.05 (12.87–47.2) | 100* | 19.18 | 19.52 | 3.20 | 3.33 |
| | I-BRD9 | 1.29 NC | 4.13 NC | 24.00NC | 28.50 (13.66–43.33) | 31.43 (31.38–31.48) | 24.44 | 7.61 | 1.31 | 1.10 |
| | SGC CBP30 | 3.19 NC | 4.99 NC | 31.22 (19.11–43.32) | 33.49 (29.61–37.36) | 50** | 15.69 | 10.03 | 1.6 | 1.49 |
| | NVS-CECR2-1 | 1.81 NC | 2.5 NC | 5.21 (0.86–9.55) | 2.90 (2.25–3.55) | 9.03 (6.60–11.46) | 4.98 | 3.60 | 1.73 | 3.11 |
| | JQ1 | 4.72 NC | 3.04 NC | 22.84 (11.87–33.81) | 18.27 (5.34–31.20) | 16.78 (15.95–17.60) | 3.56 | 5.53 | 0.73 | 0.92 |
| WRITERS | A-196 | 1.39 NC | 4.78 NC | 15.96 (8.57–23.36) | 22.29 (12.40–32.18) | 50** NC | 36.00 | 10.45 | 3.13 | 2.24 |
| | MS023 | 3.92 NC | 6.45 NC | 26.26 NC | 26.16 (5.30–47.01) | 16.25 (13.62–18.89) | 4.14 | 2.52 | 0.62 | 0.62 |
| | LLY-507 | 2.88 NC | 2.81 NC | 7.59 (4.79–10.40) | 9.32 (2.77–15.87) | 23.77 (13.9–33.7) | 8.24 | 8.47 | 3.13 | 2.55 |
| | BAY-598 | 4.81 NC | *** | 16.54 NC | 22.35NC | 50** NC | 10.40 | *** | 3.02 | 2.24 |
| | GSK343 | 3.17 NC | *** | 25.29 (23.99–26.59) | 25.04 (23.76–26.32) | 29.23 (23.58–34.87) | 9.24 | *** | 1.16 | 1.17 |
| | UNC1999 | 4.2 NC | 2.82 NC | 16.08 (2.91–29.25) | 13.53 (6.69–20.37) | 31.41 (31.4–31.4) | 7.48 | 11.16 | 1.95 | 2.32 |
| ERASERS | GSK484 | 1.18 NC | 4.94 NC | 23.11 (11.58–34.63) | 21.84 (13.04–30.64) | 31.43 (31.4–31.5) | 26.75 | 6.36 | 1.36 | 1.44 |
| | GSK-J4 | 5.37 NC | 6.04 NC | 11.39 (4.11–18.67) | 3.92 NC | 100* NC | 18.62 | 16.55 | 8.78 | 25.49 |
| | LAQ824 | 2.22 NC | 2.45 NC | 16.46 NC | 25 NC | 15.64 (13.88–17.41) | 7.04 | 6.40 | 0.95 | 0.63 |

As described in the Materials and Methods, two-fold compound titrations were performed for schistosomula (10 µM– 0.625 µM) and adult worms (50 µM– 6.25 µM or 50 µM– 0.05 µM) to calculate EC$_{50}$ values. Schistosomes and HepG2 cells were co-cultured with compounds at 37˚C in a humidified environment containing 5% $CO_2$ for 72 h and 20 h respectively.

Z´ values obtained from the schistosomula titrations were 0.44 for phenotype and 0.41 for motility.

* Cell cytotoxicity not observed at highest dose performed, therefore these compounds have a CC$_{50}$ of > 100 µM. Selectivity index calculations for these compounds were calculated with a CC$_{50}$ value set at 100 µM.

** CC$_{50}$ was not obtainable due to inaccurate slopes generated due to the requirement of further titrations > 100µM. Therefore, the predicted CC$_{50}$ is estimated to be > 50 µM. All selectivity indices for these compounds were calculated with a CC$_{50}$ of 50 µM.

*** EC$_{50}$ and selectivity indices could not be calculated due to the requirement of further compound titrations

NC—EC$_{50}$, CC$_{50}$ or 95% confidence intervals could not be calculated due to limited number of titration points, or in the case of schistosomula, due to only single replicates being performed.

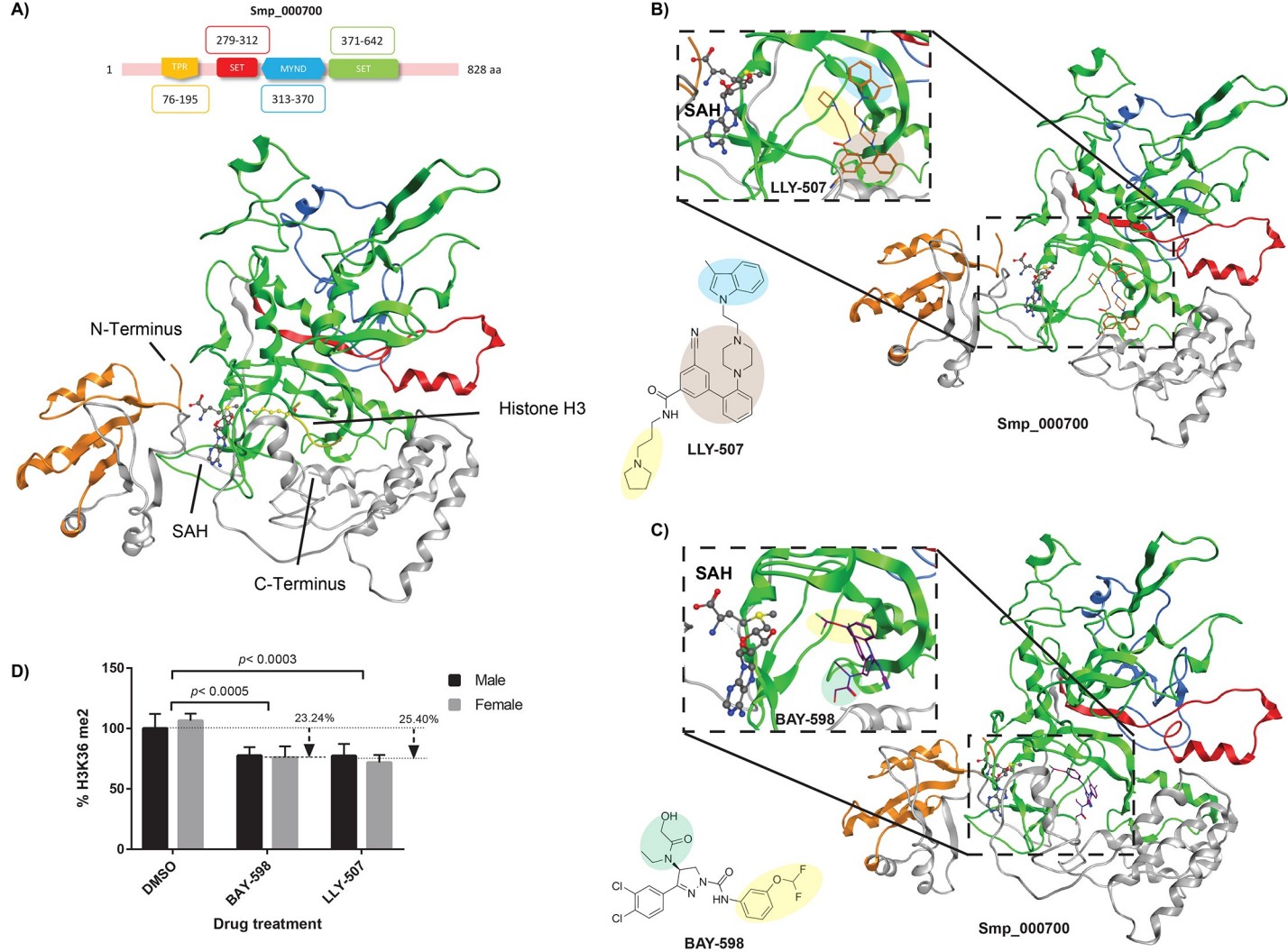

**Fig 2. The HMT inhibitors LLY-507 and BAY-598 significantly reduce adult worm H3K36me2 levels.** Homology modeling of Smp_000700, using *H. sapiens* SMYD3 (PDB 5EX3) as the template, was performed according to the Materials and Methods. **(A)** Domain architecture and homology model of Smp_000700 showing the tetratricopeptide repeat (TPR, gold, AA 76–195), the SET N-terminal domain (red, AA 279–312), the myeloid, nervy and DEAF-1 domain (MYND, blue, AA 313–370) and the SET C-terminal domain (green, AA 371–642). SAH (S-adenosyl homocysteine) and histone H3 are indicated. **(B)** Predicted binding of LLY-507 to Smp_000700 substrate binding pocket. **(C)** Predicted binding of BAY-598 to Smp_000700 substrate binding pocket. **(D)** Adult schistosome pairs (21 pairs/biological replicate; n = 3; 63 worm pairs in total) were co-cultured for 72 h in a sub-lethal concentration of LLY-507 (6.25 μM) or BAY-598 (25 μM) in 0.625% DMSO. After co-cultivation, schistosomes were separated by sex (males and females), histones extracted and total levels of H3K36me2 quantified by ELISA according to Methods.

compounds are likely acting on one of these SmSMYDs (best estimate is Smp_000700 or Smp_342100, Table 1). As *smp_000700* is more consistently expressed in 0–72 h schistosomula compared to *smp_342100* [34], we, therefore, chose to specifically examine the putative binding of LLY-507/BAY-598 to a homology model of Smp_000700 (built from HsSMYD3, PDB 5EX3) (Fig 2A). Initial assessment of the homology model by Ramachandran plot analysis, ProSA-web, ERRAT and Verify3D indicated a high-quality structure (S3 Fig) surpassing agreeable standards defined in the literature [60–63]. Together, these tools verified that Smp_000700 adopted a similar three-dimensional arrangement to that of the structurally-solved human template and was suitable for predicting the binding organization of LLY-507 and BAY-598.

Smp_000700's SET domain contains a series of β strands, which fold into three discrete sheets around a unique knot-like structure. In this structure, the C-terminal SET domain threads though a loop region, which is formed by a hydrogen bond between two segments of the protein chain [78]. The MYND domain is a putative zinc-finger motif, mainly involved in proline-rich protein interactions, and defines a distinctive class of histone writer called the SMYD proteins [79, 80]. The N-terminal SET domain is flanked by an additional domain known as tetracopeptide repeat domain (TPR). Smp_000700 contains two distinct binding sites within its SET domain: one for the substrate (histone protein, H3) and one for the demethylated cofactor (SAH). These two pockets are located on opposite faces of the SET domain and are connected by a deep channel running though the SET domain core. This architecture allows the side chain of the substrate (H3K36) to be close to the cofactor, which facilitates the transfer of a methyl group.

From the analysis of LLY-507 docking to Smp_000700, this compound adopted a similar binding confirmation to what was previously reported for HsSMYD2 bound to LLY-507 (deposited under the code 4WUY [37]). Specifically, LLY-507 is predicted to bind the substrate binding pocket with the pyrrolidine group (highlighted in yellow, Fig 2B) extending into the lysine channel (Phe184, Tyr240 and Tyr258 of HsSMYD2; His554, Tyr642 and Tyr662 of Smp_000700; S4 Fig, panels A and B) and connecting to the cofactor (represented here as SAM for HsSMYD2 and the product SAH, after donating a methyl group from SAM for Smp_000700) binding pocket. Furthermore, the nitrogen atom of LLY-507's pyrrolidine ring is predicted to form a hydrogen bond with Smp_000700 Ser555 (S4 Fig, panel B). The benzonitrile group (as well as the piperazine; both highlighted in brown, Fig 2B) binds to the periphery of the substrate binding pocket (Thr185 for HsSMYD2 and Gly643 for Smp_000700; S4 Fig, panels A and B). This interaction putatively prevents the natural substrate (histone or other target protein) from binding to Smp_000700 and supports LLY-507's mechanism of action as a substrate competitive inhibitor. Finally, this model suggests that the indole group of LLY-507 stacks into an accessory pocket of Smp_000700, just flanking the substrate binding pocket (area in blue, Fig 2B; S4 Fig, panel B). A similar indole confirmation is also observed for HsSMYD2/LLY-507 interactions (S4 Fig, panel A)

Regarding BAY-598, the docking simulation revealed a good occupation of Smp_000700's substrate binding site (Fig 2C). However, in contrast to the binding mode of this compound with the human target (HsSMYD2) where the lysine binding channel is occupied by the 3, 4-chloro phenyl residue involved in π-stacking interactions with two aromatic residues (Phe184 and Tyr240, S4 Fig, panel C) [38], this particular BAY-598 feature is predicted to bind to an adjacent hydrophobic pocket of Smp_000700 (S4 Fig, panel D). Furthermore, the 3 - (difluoromethoxy) phenyl ring of BAY-598 (in yellow, Fig 2C) is predicted to insert into the channel interconnecting the two binding pockets (substrate and cofactor) of Smp_000700 with the difluoromethoxy group pointing toward the cofactor (in this case the SAM demethylated derivative, SAH). Two Smp_000700 aromatic amino acid residues (His554 and Tyr642, S4 Fig, panel D) are likely essential for positioning BAY-598's hydroxyacetamide substituent (highlighted in green, Fig 2C) in the lysine binding channel with support from two hydrogen bonds donated by Asn641 (S4 Fig, panel D). These hydrogen bonds are donated by Thr185 in HsSMYD3 (S4 Fig, panel C) as previously shown [38]. The variation in predicted binding modes of LLY-507 and BAY-598 to Smp_000700 (i.e. LLY-507 adopting a confirmation analogous to the human homolog; BAY-598 binding to an adjacent hydrophobic pocket) may explain the difference in anti-schistosomal activity observed (LLY-507 > BAY-598). Further SAR could be helpful in providing answers to the differential activity of these two putative Smp_000700 interactors.

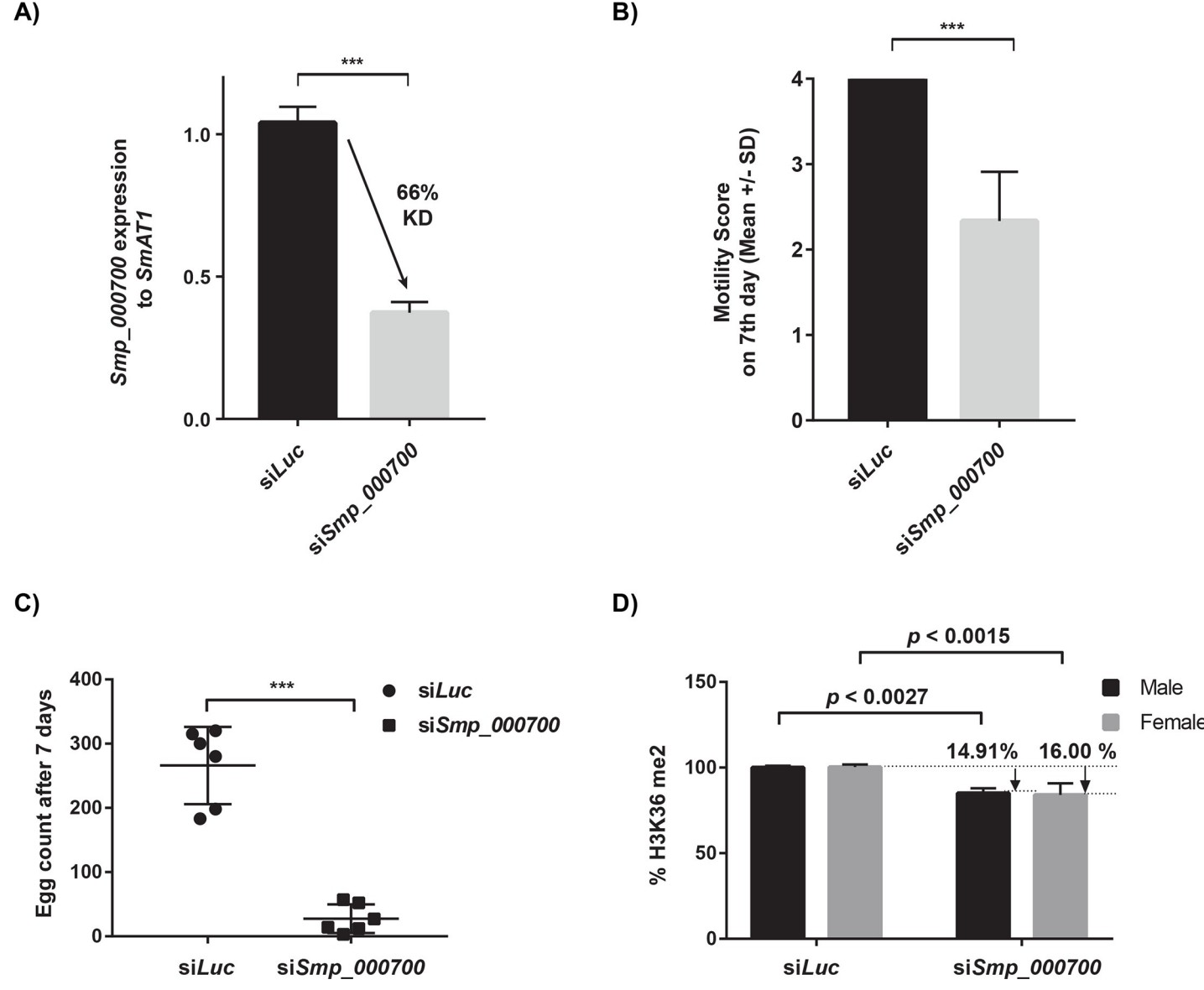

**Fig 3. Schistosome motility, egg production and H3K36me2 are regulated by Smp_000700.** RNAi of adult schistosome pairs (21 worm pairs/biological replicate; n = 3 replicates) using siRNAs directed against *smp_000700* and *luc* was performed as described in the Materials and Methods. **(A)** qRT-PCR analysis of *smp_000700* RNA levels in si*Luc* vs si*Smp_000700* treated worms at 48 h. **(B)** Quantification of adult schistosome worm motility at 168 h. **(C)** Enumeration of *in vitro* laid egg (IVLE) production at 168 h. **(D)** Detection of H3K36me2 in adult schistosome nuclear extracts at 168 h. Statistical significance is indicated (Student's t test, two tailed, unequal variance). *** represents *p* < 0.001.

The homology model and *in silico* docking analyses suggested that both LLY-507 and BAY-598 could inhibit the protein lysine methyltransferase activity of Smp_000700. Therefore, as H3K36 is a specific target of SMYD2 dimethylation (me2) [81], we specifically examined this epitope (H3K36me2) in histone preparations derived from adult schistosome pairs incubated with sub-lethal concentrations of LLY-507 (6.25 µM) or BAY-598 (25 µM) (Fig 2D). Here, a significant reduction in the amount of H3K36me2 was detected in drug-treated parasites compared to controls (DMSO) with LLY-507 slightly more effective than BAY-598 (25% vs 23%) in mediating this activity. LLY-507 and BAY-598 mediated reductions in H3K36me2 were equivalent between the sexes. Together, these data provided evidence that both SMYD2

inhibitors affected H3K36me2 in schistosomes presumably via their inhibitory activity on Smp_000700. To verify this contention, RNAi of Smp_000700 was subsequently performed on adult schistosome pairs (Fig 3).

Here, short interfering RNAs (siRNAs) targeting *smp_007000* (si*Smp_000700*) significantly reduced mRNA abundance by 66% when compared to si*Luc* controls (Fig 3A). This knockdown was also associated with a significant reduction in adult worm motility as well as egg production defects at 7 days post-treatment (Fig 3B and 3C). When H3K36me2 was examined, knockdown of *smp_000700* led to a significant reduction in this epitope in both male and female schistosomes when compared to the controls (Fig 3D). This functional genomics data broadly recapitulates the drug studies using LLY-507/BAY-598 and provides supporting evidence that Smp_000700 is an active SMYD protein lysine methyltransferase required for schistosome motility, egg production and H3K36me2. Adult worm [34] and gonad (ovaries > testes) [82] expression of *smp_000700* further provides a spatial temporal context as to how inhibition (chemical or functional genomics) of this gene product's activity may negatively affect egg-production.

Amongst the EPs/EIs studied in this investigation, GSK-J4 demonstrated the most selective activity on schistosomes (schistosomula and adults, Table 2). This EP is a cell-permeable prodrug (likely processed into GSK-J1 by intracellular schistosome esterases [39]) that is predicted to target a schistosome JMJD3 homolog (Smp_034000, Table 1) responsible for Fe (II) and 2-oxoglutarate-dependent demethylase activities. As IVLE production defects were consistently observed in adult worm screens (even at drug concentrations where worm motility was unaffected), we initiated an extensive titration (50 μM—0.05 μM) of GSK-J4 on adult schistosome pairs (Fig 4).

A direct comparison of GSK-J4 vs GSK-J1 (the cell impermeable parent of GSK-J4 [39]) on adult schistosome motility and egg production revealed the importance of an ethyl ester modification (found in GSK-J4, but not GSK-J1) in these *in vitro* phenotypic metrics (Fig 4A and 4B). For example, whereas GSK-J4 at 3.13 μM and 6.25 μM led to adult worm motility and IVLE deficiencies, GSK-J1 at these concentrations did not substantially alter these phenotypes when compared to DMSO controls. LSCM of IVLEs derived from schistosome cultures co-incubated with GSK-J4 (0.2 μM) vs GSK-J1 (6.25 μM) revealed further details about these differentially membrane permeable EPs (Fig 4C, 4D, 4E and S5 Fig). Of those phenotypically normal IVLEs found in GSK-J4 and GSK-J1 cultures at these concentrations, there was no difference in surface autofluorescence (Fig 4C and S5 Fig) or overall egg volume (Fig 4D and S5 Fig) measurements. However, when the number of vitellocytes was quantified, GSK-J4 significantly inhibited the packaging of this critical cell population into IVLEs whereas GSK-J1 did not (Fig 4C, Fig 4E and S5 Fig). Together, these data provided evidence that the specific anthelmintic activity of a JMJD3 EP is directly related to cellular (or in this case, schistosome) permeability. Alongside the results obtained with both LLY-507 and BAY-598, these findings additionally supported the important role of histone (protein) methylation/demethylation processes in schistosome development (Fig 1, Table 2) and adult worm phenotypes (Table 2, Figs 3 and 4).

## Conclusions

The active compounds identified here from within the SGC epigenetic probe/epigenetic inhibitor collection represent exciting new starting points for the pursuit of next-generation anti-schistosomals. As many of these compounds are currently being explored for the treatment of non-communicable diseases in humans [83], their direct repositioning as schistosomiasis control agents could be rapid (depending upon medicinal chemistry improvements in selectivity).

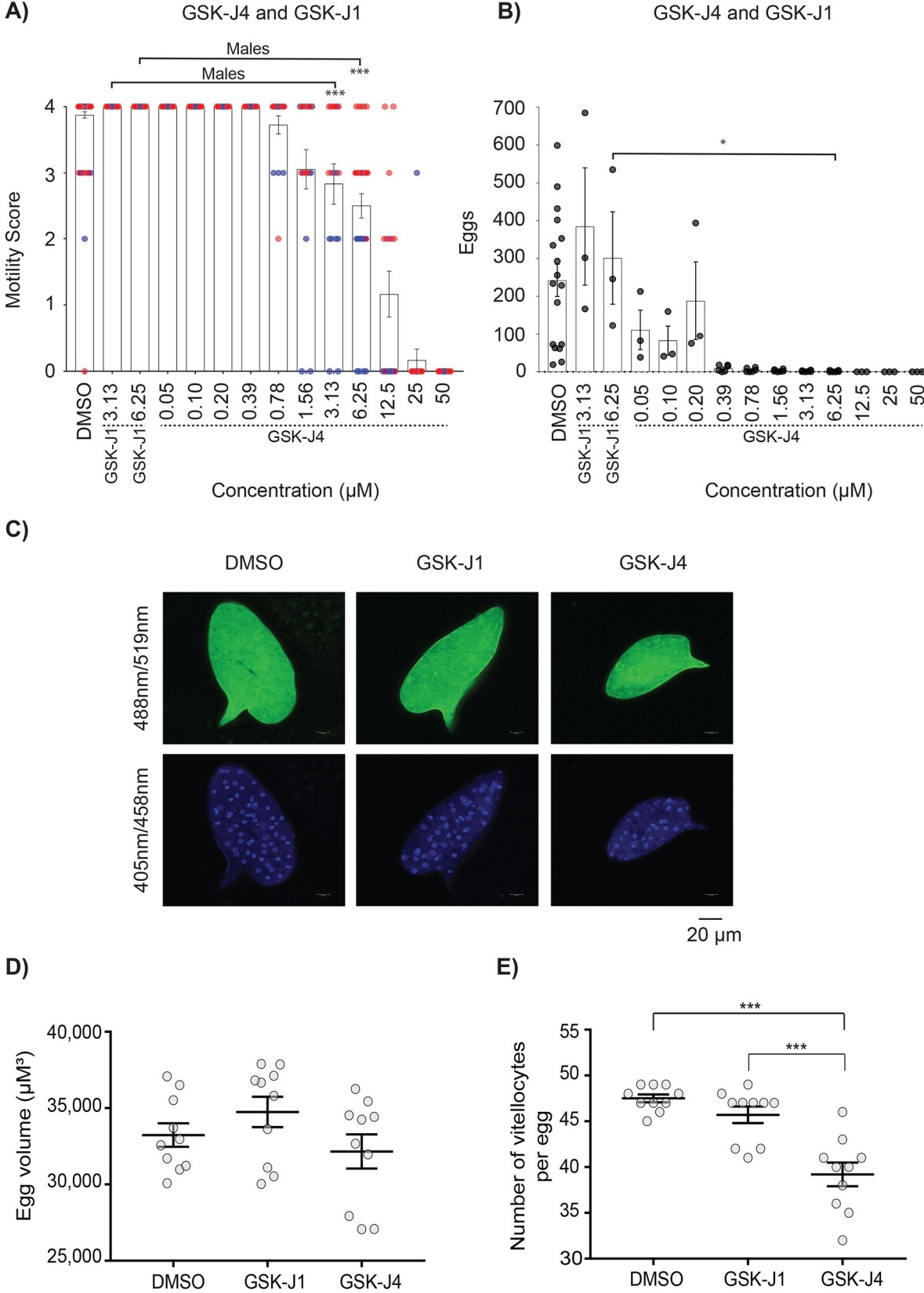

**Fig 4. The cell-permeable JMJD3 inhibitor GSK-J4, but not cell impermeable GSK-J1, significantly affects IVLE production and vitellocyte packaging.** Adult schistosome pairs (3 pairs/well; n = 3 or 6) were co-cultured for 72 h in GSK-J4 (50 μM– 0.05 μM), GSK-J1 (6.25 μM or 3.13 μM) or DMSO (0.625%) as described in the Materials and Methods. **(A)** Adult worm motility scores (red circles–males; blue circles–females). **(B)** Number of IVLEs produced. **(C)** Representative IVLE phenotypes (autofluorescence; Ex = 488 nm, Em = 519 nm and DAPI; Ex = 405 nm, Em = 458 nm) from schistosome pairs co-cultivated in GSK-J4 (0.2 μM), GSK-J1 (6.25 μM) or 0.625% DMSO for 72 h. **(D)** Quantification of egg volumes between treatment groups (n = 10 per group; GSK-J4–0.2 μM, GSK-J1–6.25 μM). **(E)** Number of vitellocytes per egg (n = 10 per group) between treatment groups (GSK-J4–0.2 μM, GSK-J1–6.25 μM). *, $p < 0.05$; ***, $p < 0.001$.

Specifically, our collective results suggest that compounds (LLY-507, BAY-598, GSK-J4) affecting protein methylation homeostasis in schistosomes are amongst some of the most potent agents from within the tested SGC collection. Therefore, and in agreement with other studies [17, 34, 36, 66], members of the schistosome protein methyltransferase/demethylase families should be considered validated anthelmintic targets for progression.

## Supporting information

**S1 Table. Structural Genomics Consortium (SGC) epigenetic probes (EPs) and epigenetic inhibitors (EIs) used in this study.**
(XLSX)

**S1 Fig. Activity of SGC chemotype-matched negative control compounds on schistosomula phenotype and motility.** Chemotype-matched control compounds of EPs UNC1999 (UNC2400), A-196 (A-197), BAY-598 (BAY-369) and GSK-J4 (GSK-J1 and GSKJ5) as well as Auranofin were screened against *S. mansoni* schistosomula at 10 μM for 72 h using the Robo-worm platform as described in the Methods. **(A)** Compounds were listed as a hit if they fell within both phenotype and motility cut-off thresholds (-0.15 and -0.35, respectively) [50]. **(B)** Representative images of schistosomula phenotypes induced by co-cultivation with these EPs and chemotype matched controls, compared to schistosomula co-cultivated with media only, DMSO (0.625%) and Auranofin.
(PDF)

**S2 Fig. Phylogenetic analysis of *S. mansoni* and *H. sapiens* SMYD protein members.** The phylogeny outlined in the tree is derived from multiple sequence alignment of the SET domain of 3 SmSMYDs (Smp_000700, Smp_121610 and Smp_342100) and 5 HsSMYDs (SMYD1-5). The consensus tree is constructed in MEGA using the neighbor joining method. An unrooted dendogram represents the bootstrap analysis of the HsSMYD and SmSMYD members accomplished using 1000 iterations. The taxa name (sequence name) is reported at the tip of each branch and the bootstrap value (supportive value) is indicated for each node. The branch length is proportional to the distance calculated between the various SMYD family members with the scale reported as reference at the bottom of the dendogram.
(PDF)

**S3 Fig. Catalytic domain of Smp_000700 homology model evaluation. (A)** Ramachandran plot showing the dihedral Psi and Phi angles of amino acid residues within the catalytic domain of Smp_000700 (SET domain, 413 aa in length). This analysis illustrates that 98.6% of modelled residues satisfy stereochemical parameters. In fact, various residues lie in the general favoured regions (black symbols in blue and orange areas on the graph) and the allowed regions (orange symbols in blue and orange areas on the graph). Very few residues lie within the white field, which represents disallowed regions. **(B)** Z-score of Smp_000700's SET domain provided by ProSA-web. The black dot (highlighted by the arrow) represents this Z-score (-7.11) in relation to all protein chains in PDB determined by X-ray crystallography (light blue area) or NMR spectroscopy (dark blue area) with respect to their length (x-axis representing

the protein length in terms of number of residues). Our model is located within the space occupied by protein structures solved by X-ray crystallography. (**C**) Smp_000700 model quality (over SET domain) assessed by the protein verification tool ERRAT. Error values are plotted as a function of a sliding 9-residue window; poorly supported model residues (highest bars on the Errat Plot) are coloured red (rejected at 99% confidence level or above) or yellow (between 95% and 99% confidence levels). Regions of the structure not rejected are shown in green. Overall ERRAT score of Smp_000700's SET domain is 88.15%. (**D**) Evaluation of Smp_000700 homology model (SET domain) was additionally conducted by Verify3D, which determines the compatibility of an atomic tertiary model (3D) from its own primary amino acid sequence (1D). As a result, 81.90% of the SET domain residues have a good score ($> = 0.2$) compatible with the formation of a stable 3D structure. (**E**) Quality structure assessment summary of Smp_000700 homology model (SET domain) and the corresponding human template (SMYD3, PDB ID: 5EX3). This final table summarises the results of the structural validation of both structures compared to the expected values for the four tools.
(PDF)

**S4 Fig. Binding of LLY-507 and BAY-598 to HsSMYD2 and Smp_000700.** Views of the co-crystal structure of LLY-507 with HsSMYD2 (PDB ID: 4WUY; Panel **A**) compared to the predicted binding of LLY-507 with the homology model of Smp_000700 (Panel **B**). Similar comparisons were made between the co-crystal structure of BAY-598 with HsSMYD2 (PDB ID: 5ARG; Panel **C**) and the homology model of Smp_000700 (Panel **D**). SAM (S-adenosyl methionine, for HsSMYD2), SAH (S-adenosyl homocysteine, for Smp_000700) and the compound structures are shown as ball-and-stick diagrams, coloured by atom type: grey for carbons, red for oxygen, blue for nitrogen. The human and parasite proteins are shown as green and blue ribbon, respectively. Residues interacting with the compounds are shown in stick mode and the relative numeration refers to their positions on the full-length protein sequence. For clarity, hydrogens, small portion of the ribbon and protein side chains and backbones (except for the highlighted residues) are not shown.
(PDF)

**S5 Fig. Representative egg phenotypes derived from adult worm cultures co-incubated with GSK-J1 and GSK-J4.** Representative IVLE phenotypes (GFP = eggshell surface autofluorescence; Ex = 488 nm, Em = 519 nm and DAPI = vitellocytes; Ex = 405 nm, Em = 458 nm) from schistosome pairs co-cultivated in GSK-J4 (0.2 µM), GSK-J1 (6.25 µM) or 0.625% DMSO for 72 h.
(TIF)

## Acknowledgments

We thank past and current members of the Hoffmann group for contributions to *S. mansoni* lifecycle maintenance, Dr Jaroslaw Tomczak (Informatics Unlimited, Ltd) for assisting in Roboworm upkeep and the SGC, the provider of compounds in this study and a registered charity (number 1097737).

## Author Contributions

**Conceptualization:** Karl F. Hoffmann.

**Data curation:** Kezia C. L. Whatley, Gilda Padalino, Helen Whiteland, Benjamin J. Hulme.

**Formal analysis:** Kezia C. L. Whatley, Gilda Padalino, Benjamin J. Hulme.

**Funding acquisition:** Andrea Brancale, Karl F. Hoffmann.

**Investigation:** Kezia C. L. Whatley, Gilda Padalino.

**Methodology:** Kezia C. L. Whatley, Benjamin J. Hulme.

**Project administration:** Karl F. Hoffmann.

**Resources:** Salvatore Ferla, Andrea Brancale, Karl F. Hoffmann.

**Supervision:** Helen Whiteland, Kathrin K. Geyer, Iain W. Chalmers, Josephine Forde-Thomas, Salvatore Ferla, Andrea Brancale, Karl F. Hoffmann.

**Validation:** Gilda Padalino.

**Visualization:** Kezia C. L. Whatley, Gilda Padalino, Helen Whiteland, Benjamin J. Hulme.

**Writing – original draft:** Kezia C. L. Whatley, Gilda Padalino, Karl F. Hoffmann.

**Writing – review & editing:** Kezia C. L. Whatley, Gilda Padalino, Helen Whiteland, Kathrin K. Geyer, Benjamin J. Hulme, Iain W. Chalmers, Josephine Forde-Thomas, Salvatore Ferla, Andrea Brancale, Karl F. Hoffmann.

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
