## [Decision Letter · Decision Letter 0]

31 Aug 2019

Dear Prof. Hoffmann:

Thank you very much for submitting your manuscript "The repositioning of epigenetic probes/inhibitors identifies new anti-schistosomal lead compounds and chemotherapeutic targets" (#PNTD-D-19-01335) for review by PLOS Neglected Tropical Diseases. Your manuscript was fully evaluated at the editorial level and by independent peer reviewers. The reviewers appreciated the attention to an important problem, but raised a substantial concerns about the manuscript as it currently stands. This issue must be addressed before we would be willing to consider a revised version of your study. We cannot, of course, promise publication at that time.

We therefore ask you to modify the manuscript according to the review recommendations before we can consider your manuscript for acceptance. Your revisions should address the specific points made by each reviewer. 

When you are ready to resubmit, please be prepared to upload the following:

(1) A letter containing a detailed list of your responses to the review comments and a description of the changes you have made in the manuscript.

(2) Two versions of the manuscript: one with either highlights or tracked changes denoting where the text has been changed (uploaded as a "Revised Article with Changes Highlighted" file); the other a clean version (uploaded as the article file).

(3) If available, a striking still image (a new image if one is available or an existing one from within your manuscript). If your manuscript is accepted for publication, this image may be featured on our website. Images should ideally be high resolution, eye-catching, single panel images; where one is available, please use 'add file' at the time of resubmission and select 'striking image' as the file type. 

Please provide a short caption, including credits, uploaded as a separate "Other" file. If your image is from someone other than yourself, please ensure that the artist has read and agreed to the terms and conditions of the Creative Commons Attribution License at http://journals.plos.org/plosntds/s/content-license (NOTE: we cannot publish copyrighted images). 

(4) If applicable, we encourage you to add a list of accession numbers/ID numbers for genes and proteins mentioned in the text (these should be listed as a paragraph at the end of the manuscript). You can supply accession numbers for any database, so long as the database is publicly accessible and stable. Examples include LocusLink and SwissProt.

(5) To enhance the reproducibility of your results, we recommend that you deposit your laboratory protocols in protocols.io, where a protocol can be assigned its own identifier (DOI) such that it can be cited independently in the future. For instructions see http://journals.plos.org/plosntds/s/submission-guidelines#loc-methods

While revising your submission, please upload your figure files to the Preflight Analysis and Conversion Engine (PACE) digital diagnostic tool, https://pacev2.apexcovantage.com/ PACE helps ensure that figures meet PLOS requirements. To use PACE, you must first register as a user. Then, login and navigate to the UPLOAD tab, where you will find detailed instructions on how to use the tool. If you encounter any issues or have any questions when using PACE, please email us at figures@plos.org.

We hope to receive your revised manuscript by Oct 30 2019 11:59PM. If you anticipate any delay in its return, we ask that you let us know the expected resubmission date by replying to this email.

To submit a revision, go to https://www.editorialmanager.com/pntd/ and log in as an Author. You will see a menu item call Submission Needing Revision. You will find your submission record there. 

Sincerely,

Philip T. LoVerde

Guest Editor

Jennifer Keiser

Deputy Editor

Your manuscript, indicated above, has been evaluated by three referees whose comments are attached. In addition, I have reviewed both the referee’s comments and the manuscript itself. The referees provided a number of comments and suggestions for improving the manuscript. There was one major comment that is worth addressing that is "why [the] authors used such a small incubation time (20h) with cytotoxicity assays on the human cell line for calculating selectivity indexes. It's an issue since parasite assays were all performed with 72h incubation time. The authors should generate new data with 72h incubation or make a strong case justifying their choice for 20h since keeping the later should overestimated the selectivity/safety of compounds"

Reviewer's Responses to Questions

Key Review Criteria Required for Acceptance?

Methods

-Are the objectives of the study clearly articulated with a clear testable hypothesis stated?

-Is the study design appropriate to address the stated objectives?

-Is the population clearly described and appropriate for the hypothesis being tested?

-Is the sample size sufficient to ensure adequate power to address the hypothesis being tested?

-Were correct statistical analysis used to support conclusions?

-Are there concerns about ethical or regulatory requirements being met?

Reviewer #1: -Are the objectives of the study clearly articulated with a clear testable hypothesis stated?

Yes

-Is the study design appropriate to address the stated objectives?

Yes

-Is the population clearly described and appropriate for the hypothesis being tested?

NA

-Is the sample size sufficient to ensure adequate power to address the hypothesis being tested?

NA

-Were correct statistical analysis used to support conclusions?

Yes

-Are there concerns about ethical or regulatory requirements being met?

NA

Reviewer #2: Several minor details in the methods text could be clarified. 

- Line 149. What strain of Mus musculus?

- Line 188. Schistosomules were cultured with compound at 37 degrees – were they also imaged at 37 or at room temperature?

- Line 189. Reference 50 (Paveley et al., 2012) is mentioned regarding phenotype and motility. However, some details on this package could be clarified. Is this package publically available? What was the rational for the hit thresholds (dotted line figure 1A and 1B)?

- For RNAi, what concetration of siRNA duplexes were used?

- Line 263. How many worms sample input are used in this assay?

Reviewer #3: Methods are appropriate for the study

Results

-Does the analysis presented match the analysis plan?

-Are the results clearly and completely presented?

-Are the figures (Tables, Images) of sufficient quality for clarity?

Reviewer #1: -Does the analysis presented match the analysis plan?

Yes

-Are the results clearly and completely presented?

Mostly.

-Are the figures (Tables, Images) of sufficient quality for clarity?

Yes.

Reviewer #2: The results are clearly presented. As mentioned above in the methods, a description of the hit threshold scores shown in Figure 1A and 1B could be elaborated upon. Either on the legend, or in the text around line 361-364 when these cutoffs are mentioned.

For table 2, add in either the table or the legend a description of the assay used to generate the IC50 data for adult males and females. Also, in the text around line 409 where these results are discussed.

The structures of the chemicals shown in table 2 should be included. It may not be convenient to include them in the actual table 2, but perhaps a supplemental table or in text form (for example, in SMILES format). 

While this is not essential, I am interested in whether pharmacological or genetic inhibition of histone methylation effects mitotic activity of somatic neoblasts or germ tissues (testies, ovaries, vitellaria)? Is this the mechanism by which egg laying is arrested?

Reviewer #3: Results match the analysis plan.

Conclusions

-Are the conclusions supported by the data presented?

-Are the limitations of analysis clearly described?

-Do the authors discuss how these data can be helpful to advance our understanding of the topic under study?

-Is public health relevance addressed?

Reviewer #1: -Are the conclusions supported by the data presented?

Generally, yes.

-Are the limitations of analysis clearly described?

Mostly.

-Do the authors discuss how these data can be helpful to advance our understanding of the topic under study?

Yes.

-Is public health relevance addressed?

Yes.

Reviewer #2: The conclusions are well supported by the data. Some discussion could elaborate the repurposing of epigenetic probes / inhibitor compounds. As mentioned in the introduction, praziquantel is not effective against juvenile life stages (~4 week, liver stage immature worms). Granted, ex vivo assays on these worms are more challenging – but is anything known about the expression of targets such as Smp_007000 across the schistosome life cycle? Given the effects on egg laying, it would be interesting whether these are differentially expressed in males and females or in the gonads (PMID: 27499125).

Reviewer #3: Conclusions are appropriate.

Editorial and Data Presentation Modifications?

Reviewer #1: (No Response)

Reviewer #2: (No Response)

Reviewer #3: minor revision

Summary and General Comments

Reviewer #1: The authors describe the anti-schistosomal activities of compounds that are known modulators/inhibitors of human molecular targets involved in epigenetic processes. The deck of 37 compounds was obtained from the Structural Genomics Consortium and tested against schistosomula and adult schistosome lifecycle stages. The study combines several approaches to support/validate the main hits from the initial screenings: in silico modelling, RNAi-mediated functional genomics, enzyme assays and confocal microscopy. 

This is an important work with interesting findings to schistosomal drug discovery targeting epigenetic processes. Hence, it should be highly appreciated by the journal readership. I list below several minor points to be corrected but the only major point to be addressed is why authors used such a small incubation time (20h) with cytotoxicity assays on the human cell line for calculating selectivity indexes. It's an issue since parasite assays were all performed with 72h incubation time. The authors should generate new data with 72h incubation or make a strong case justifying their choice for 20h since keeping the later should overestimated the selectivity/safety of compounds.

Minor points:

- Methods:

1- Compound acquisition, storage and handling: Please indicate where appropriate that compounds structures can be found at Table S1

2- Homology modelling and epi-drug docking: 

2a - Provide (as supplementary material) evidence for model validation and quality, having special attention to any issues involved in the proposed docking site.

2b - Are all the EP/Eis know to be competitive with the substrates? if so, authors should state it earlier.

- Results:

Lines 311-313: Kd, Ki or IC50? the term potency may be misleading since it's usually expressed as an inverse proportional term. Then, higher potency may actually mean < 100nm.="" lines="" _348-3493a_="" figure="" 1="" _legend3a_="" please="" indicate="" if="" dashes="" crossing="" vertical="" connecting="" each="" replicate="" mean="" averages="" or="" medians.="" line="" _3853a_="" _please2c_="" complement="" s1="" by="" including="" chemical="" structures="" for="" the="" compounds.="" _386-3873a_="" authors="" contend="" fact="" that="" none="" of="" 386="" these="" chemotype="" controls="" demonstrated="" anti-schistosomula="" activity="" suggests="" specificity="" hits="" s.="" mansoni="" homologue="" is="" similar="" to="" found="" original="" human="" target="" but="" other="" interpretation="" would="" be="" they="" lack="" proper="" sar="" targets.="" it="" should="" discussed.="" table="" _23a_="" all="" ec50="" and="" cc50="" accompanied="" standard="" errors="" _9525_="" confidence="" intervals.="" also="" incubation="" time="" below="" headings.="" in="" _footnote2c_="" define="" _22_accurate22_.="" _413-4153a_="" are="" there="" any="" discussed="" compounds="" lly-507="" _bay-5983f_="" _4443a_="" _22_strong22_="" _22_close22_="" _463-4783a_="" make="" their="" conclusions="" lighter="" using="" expressions="" like="" _22_is="" predicted="" _bind...22_="" _22_predicted="" interaction="" may="" _prevent...22_="" _5233a_="" clarify="" why="" gsk-j4="" a="" prodrug.="" how="" drug="" _processed3f_="" _typos3a_="" _713a_="" theat="" threat="" _1273a_="" _22_cultured="" _this22_="" with="" _4523a_="" _22_ceb2_="" _stands22_="" _strands22_="" reviewer="" _23_23a_="" this="" study="" screened="" panel="" known="" epigenetic="" against="" schistosomules="" then="" proceeded="" active="" more="" detail="" on="" adult="" parasites="" _e28093_="" profiling="" phenotypic="" outcomes="" such="" as="" motility="" egg="" _laying2c_="" well="" assaying="" histone="" methylation.="" pharmacological="" experiments="" were="" controlled="" negative="" further="" validated="" rnai="" data="" showing="" knockdown="" putative="" methyltransferase="" phenocopies="" treatment.="" convincing="" controlled.="" laying="" phenotype="" particular="" _promising2c_="" since="" pathology="" associated="" disease="" driven="" parasite="" eggs.="" several="" suggestions="" clarification="" methods="" results="" have="" been="" outlined.="" _23_33a_="" manuscript="" written="" will="" interest="" pntd="" readers.="" minor="" revisions="" suggested.="" abstract="" author="" summary="" _numerous2c_="" undefined="" abbreviations.="" distracting="" _confusing3a_="" _482c_="" _532c_="" 76.="" _general2c_="" has="" large="" number="" _abbreviations2c_="" making="" tedious="" _read2c_="" do="" not="" addressed.="" 54="" _1525_="" something="" indicated.="" define.="" at="" places="" _e2809c_cell="" permeable="" _prodruge2809d_="" described.="" _however2c_="" _e2809c_worm="" _permeablee2809d_="" another="" matter="" general="" poorly="" understood.="" from="" overstated="" absence="" direct="" enzymatic="" evidence="" inhibition.="" was="" done="" indirectly="" h3k36me2="" levels="" _studies2c_="" which="" better="" support="" inhibitor="" _link2f_mechanism="" action.="" 311="" 313="" /> signs should be reversed.

A general comment: the reality of therapy for schistosomiasis is that compounds need to be active after a single dose, where active means worm killing. Blocking motility or egg production after extended exposure to a compound will not be adequate for novel therapies. Did the compounds tested actually _kill_ the worms?

PLOS authors have the option to publish the peer review history of their article (what does this mean?). If published, this will include your full peer review and any attached files.

Do you want your identity to be public for this peer review? For information about this choice, including consent withdrawal, please see our Privacy Policy.

Reviewer #1: Yes: Floriano Paes Silva-Jr

Reviewer #2: No

Reviewer #3: No

---

## [Editor Report · Decision Letter 1]

30 Oct 2019

Dear Prof. Hoffmann,

We are pleased to inform you that your manuscript, "The repositioning of epigenetic probes/inhibitors identifies new anti-schistosomal lead compounds and chemotherapeutic targets", has been editorially accepted for publication at PLOS Neglected Tropical Diseases.

Before your manuscript can be formally accepted and sent to production you will need to complete our formatting changes, which you will receive in a follow up email. Please note: your manuscript will not be scheduled for publication until you have made the required changes.

IMPORTANT NOTES

* Copyediting and Author Proofs: To ensure prompt publication, your manuscript will NOT be subject to detailed copyediting and you will NOT receive a typeset proof for review. The corresponding author will have one final opportunity to correct any errors when sent the requests mentioned above. Please review this version of your manuscript for any errors.

* If you or your institution will be preparing press materials for this manuscript, please inform our press team in advance at plosntds@plos.org. If you need to know your paper's publication date for media purposes, you must coordinate with our press team, and your manuscript will remain under a strict press embargo until the publication date and time. PLOS NTDs may choose to issue a press release for your article. If there is anything that the journal should know, please get in touch.

*Now that your manuscript has been provisionally accepted, please log into EM and update your profile. Go to http://www.editorialmanager.com/pntd, log in, and click on the "Update My Information" link at the top of the page. Please update your user information to ensure an efficient production and billing process.

*Note to LaTeX users only - Our staff will ask you to upload a TEX file in addition to the PDF before the paper can be sent to typesetting, so please carefully review our Latex Guidelines [http://www.plosntds.org/static/latexGuidelines.action] in the meantime.

Best regards,

Philip T. LoVerde

Guest Editor

Jennifer Keiser

Deputy Editor

The authors have satisfactorily addressed the concerns of the reviewers.

---

## [Editor Report · Acceptance letter]

7 Nov 2019

Dear Prof. Hoffmann,

We are delighted to inform you that your manuscript, "The repositioning of epigenetic probes/inhibitors identifies new anti-schistosomal lead compounds and chemotherapeutic targets," has been formally accepted for publication in PLOS Neglected Tropical Diseases.

Best regards,

Serap Aksoy

Editor-in-Chief

Shaden Kamhawi

Editor-in-Chief
